

# Development of core-collections for Guizhou tea genetic resources and GWAS of leaf size using SNP developed by genotyping-by-sequencing

Suzhen Niu[1,2], Hisashi Koiwa[3], Qinfei Song[2], Dahe Qiao[1], Juan Chen[1], Degang Zhao[1], Zhengwu Chen[1], Ying Wang[4] and Tianyuan Zhang[4]

[1] Guiyang Station for DUS Testing Center of New Plant Varteties (MOA) / Institute of Tea, Guizhou Academy of Agricultural Sciences, Guiyang, China
[2] The Key Laboratory of Plant Resources Conservation and Germplasm Innovationin Mountainous Region (Ministry of Education), Institute of Agro-Bioengineering / College of Tea Science, Guizhou University, Guiyang, China
[3] Vegetable and Fruit Improvement Center, Department of Horticultural Sciences, Molecular and Environmental Plant Sciences Program, Texas A&M University, College Station, Texas, USA
[4] Wuhan Benagen Tech Solutions Company Limited, Wuhan, China

Corresponding authors
Degang Zhao, dgzhao@gzu.edu.cn
Zhengwu Chen, zwchentea@163.com

## ABSTRACT

An accurate depiction of the genetic relationship, the development of core collection, and genome-wide association analysis (GWAS) are key for the effective exploitation and utilization of genetic resources. Here, genotyping-by-sequencing (GBS) was used to characterize 415 tea accessions mostly collected from the Guizhou region in China. A total of 30,282 high-quality SNPs was used to estimate the genetic relationships, develop core collections, and perform GWAS. We suggest 198 and 148 accessions to represent the core set and mini-core set, which consist of 47% and 37% of the whole collection, respectively, and contain 93–95% of the total SNPs. Furthermore, the frequencies of all alleles and genotypes in the whole set were very well retained in the core set and mini-core set. The 415 accessions were clustered into 14 groups and the core and the mini-core collections contain accessions from each group, species, cultivation status and growth habit. By analyzing the significant SNP markers associated with multiple traits, nine SNPs were found to be significantly associated with four leaf size traits, namely MLL, MLW, MLA and MLSI ($P < 1.655\text{E}{-}06$). This study characterized the genetic distance and relationship of tea collections, suggested the core collections, and established an efficient GWAS analysis of GBS result.

## INTRODUCTION

Tea, coffee and cocoa are considered as the world's three best-known beverages and are also the most produced or consumed beverages worldwide (*Wambulwa et al., 2016*; *Liu et al., 2017*). To produce tea, people need to infuse processed tender shoots of the tea plant (*Camellia sinensis*) with connected species, all of which belong to *Thea* of genus *Camellia* in the family Theaceae (*Ma et al., 2018a*; *Ma et al., 2018b*; *Ma et al., 2018c*). Tea is a diploid

$(2n = 2x = 30)$ with a genome size of 3.02 Gb and is an endemic species in southwest China (*Wei et al., 2018*). Tea is highly nutritious with medicinal properties, offering a wealth of health benefits. Daily tea consumption is beneficial for reducing the risks of various cancers, diseases related to obesity, and neurological and cardiovascular dysfunctions (*Naghma & Hasan, 2013*; *Hayat et al., 2015*). The rich flavors and multiple health-promoting functions of tea are conferred by 700 bioactive compounds such as catechins, caffeine, theanine, and volatiles (*Xia et al., 2017*).

Tea has a long utilization history in the Yunnan-Guizhou Plateau and is now cultivated worldwide (*Wei et al., 2018*; *Chen, Apostolides & Chen, 2012*). In this region, the diversity of tea germplasm is well preserved with abundant wild tea plants, ancient landraces and modern landraces with different morphological characteristics due to the region's unique geology, diverse climates, plentiful rainfall and the cross-pollination nature of tea (*Niu, 2014*). In addition, due to the slow socio-economic development and land use in Guizhou Plateau, elimination of various tea species at a large spatial scale has not occurred.

The ever-growing popularity of tea products calls for new tea varieties that meet the needs of market diversification (*Liang & Shi, 2015*). Similar to other perennial woody species, tea plants have a long growth period and the genome is highly heterozygous, making breeding difficult and costly. Tea producers also face new challenges such as the sustainability of high-quality tea production, environmental change, pest invasion, and diseases. Hence, it is vital to explore the genetic basis of complex traits and identify favorable alleles for breeding new tea cultivars that can overcome these threats by marker-assisted breeding (MAB) (*Tan et al., 2016*).

Compared with other ways, association mapping is faster and more effective in dissecting the genetic basis of complex traits and identifying favorable genetic resource. The results yielded from association mapping have greatly facilitated MAB programs (*Iso-Touru et al., 2016*). Association mapping is also promising in evading limitations of linkage mapping. The difference between traditional linkage mapping which is often based on bi-parental populations and association mapping is that the latter takes advantage of the ancestral recombination in natural populations to identify loci that significantly associate with traits of interest based on the linkage disequilibrium (LD) (*Buckler & Thornsberry, 2002*; *Pace et al., 2015*). Analysis of a large number of alleles in various populations is made possible by association mapping (*Pace et al., 2015*; *Suwarno et al., 2015*; *Motilal et al., 2016*). Other advantages of GWAS compared with traditional linkage mapping (*Tan et al., 2016*; *Bali et al., 2015*) include high-resolution and providing a less time-consuming approach for developing the mapping population (*Heena et al., 2018*).

The population used for association mapping is of great importance, it must have a wide range of diversity that represents most historical recombination events (*Heena et al., 2018*). Although species diversity with abundant tea genetic collections is well preserved in the center of origin, recent tea breeding programs have not exploited the allelic diversity of many traits. As a result, characterizing tea plant collections should be the first step towards stimulating the use of genetic resources. Germplasm preservation can be costly and time-consuming, therefore, developing 'core collections' with a minimal number of tea varieties that retain the genetic diversity is a cost-effective strategy (*Frankel, 1984*; *Brown,*

*1989*; *Wang, Chen & Yang, 2011*; *Taniguchi, Kimura & Saba, 2014*; *Taniguchi, McCloskey & Ohno, 2014*; *Campoy et al., 2016*; *Ndjiondjop et al., 2017*; *Lassois et al., 2016*). In addition, 'core-collections' are useful for genetic association analysis and the identification of genomic variation (*Huggins et al., 2018*; *Zhang et al., 2018*; *Cunff et al., 2008*; *Heena et al., 2018*; *Ma et al., 2018a*; *Ma et al., 2018b*; *Ma et al., 2018c*). Criteria based on the genetic distance between accessions have been proven sueful in evaluating and creating 'core collections' (*Odong et al., 2013*; *Campoy et al., 2016*; *Ndjiondjop et al., 2017*).

Previous studies on association mapping and core-collection development mainly focused on maize (*Pace et al., 2015*; *Suwarno et al., 2015*; *Coan et al., 2018*), sunflower (*Heena et al., 2018*), wheat (*Muleta et al., 2017*), pine (*Bartholomé et al., 2017*), sorghum (*Bouchet et al., 2017*), finger millet (*Babu et al., 2018*), melon (*Hou et al., 2018*), pea (*Desgroux et al., 2017*) and cotton (*Ma et al., 2018a*; *Ma et al., 2018b*; *Ma et al., 2018c*). By contrast, little has been done on tea germplasm with high density SNPs distributed on the whole genome. GBS is a large-scale approach for identifying high density SNP markers that are suitable for association mapping. In this study, GBS was used to analyze the genetic diversity and population structure of 415 tea accessions including wild types, ancient and modern landraces from the Guizhou Plateau, and breeding varieties from Zhejiang, Fujian, Hunan and Guizhou. We aim to (1) analyze the genetic purity, distance and relationships, (2) establish and evaluate the core collections, and (3) perform GWAS on the tea population. Our findings provide a valuable resource for developing new molecular markers that can be used for MAB of tea varieties.

## MATERIAL AND METHODS

### Plant materials

A total of 415 accessions including 159 wild type and 256 cultivation type varieties (174 ancient landraces, 77 modern landraces and 5 breeding cultivars) were used for this study (Fig. S1; Table S1). Based on the classification system established by *Chen, Yu & Tong (2000)* and *Min (1992)*, 251 *C.sinensis* (L.) O. Ktze, 100 *C.tachangensis* F.C. Zhang, 59 *C. remotiserrata* Zhang, and five near *C.taliensis* W.W. Smith were identified (Table S1). Hereafter, wild varieties and their natural offspring will be referred to as "wild type", the cultivation tea varieties aged more than one hundred years old will be called "ancient landraces", and the tea garden landraces will be named "modern landraces" (Table S1). The ancient landraces, modern landraces and "breeding varieties" that have undergone artificial selection are collectively referred to as the "cultivation type".

One hundred and sixty-eight samples were collected from areas suitable for tea growth in north Guizhou (Ia) (Fig. S1), 51 samples were collected from areas suitable for tea growth in east Guizhou (Ib), 57 samples were collected from areas suitable for tea growth in south Guizhou (Ic), 83 samples were collected from areas suitable for tea growth in central Guizhou (II), 41 samples were collected from areas with a minor suitable climate for tea growth in west Guizhou (III), ten samples were collected from areas with an unsuitable climate for tea growth in west Guizhou, one cultivar was collected from Guizhou, two cultivars were collected from Fujian, one cultivar was collected from Zhejiang, and one

cultivar was collected from Hu'nan (Table S1). The samples were planted in the city of Guiyang, China. The healthy tender shoots were harvested, snap-frozen in liquid $N_2$, and kept at $-80\,°C$ until use (*Jin et al., 2018*).

## DNA extraction

Genomic DNA was extracted from the plant samples with a kit for rapid extraction of genomic DNA. DNA integrity was tested by 1% agarose gel electrophoresis; a Qubit Fluorometer was used to check the purity and measure the concentration of DNA samples (*Niu et al., 2019*).

## Sequencing of the GBS library

One-hundred nanogram genomic DNA was double digested with 5 U of *Sac* I and *Mse* I (NEB) in a 25 µl reaction containing 1 × restriction buffer. The resulting samples were ligated with the restriction fragment using the SacAD and MseAD adaptors, which contained different barcode combinations to distinguish the samples. Equal volumes of the ligated products of 12 individuals were pooled and processed using the QIA quick PCR Purification Kit (Qiagen). PCR was performed using PCR Master Mix and the PCR Primer Cocktail, which could enrich DNA fragments with the adapters. The PCR products of each mixture were pooled and separated on a 2% agarose gel by electrophoresis. Fragments of 500 to 550 bp (including the 120 bp adaptor) were recovered with the QIA quick Gel Extraction Kit (Qiagen). The average length of DNA fragment was selected for final library construction with the Agilent DNA 12,000 kit using the 2100 Bioanalyzer system. To quantify the final library, quantitative real-time PCR with a Taq Man probe was employed. Then the libraries were sequenced on the IlluminaHiSeq X Ten platform based on the paired-end 150 (PE150) strategy. Each library contains 48 samples, and clean data were then parsed into different units that exactly match the barcodes and the restriction sites at both ends (*Elshire et al., 2011*).

## SNP genotyping basing on the sequence data

Original IlluminaHiSeq X Ten reads were de-multiplexed based on the barcodes, and a custom Perl script was used to shear the barcoded sequences. Reads in which >50% of the bases have quality values ≤5 were discarded; the clean reads were mapped to the tea reference genome (http://www.plantkingdomgdb.com/tea_tree/) (*Xia et al., 2017*) using BWA-MEM (v.0.7.10) with parameters '-T 20 -k 30' (*Li, 2013*). SNPs and InDels were called using GATK (v.3.7.0).

The SNPs were filtered according to the methods described by *Hussain et al. (2017)*, *Chen et al. (2017)* and *Eltaher et al. (2018)* based on the following criteria: (1) variants must be bi-allelic SNPs, (2) "QUAL <50.0 || QD <2.0 || FS >60.0 || MQ <40.0 || Mapping Quality Rank Sum <−12.5 || Read Pos Rank Sum <−8.0" was used in Variant Filtration in GATK (V 3.7.0) to filter the SNPs, (3) SNPs with MAF <0.05 or >10% missing data were filtered out by VCFtools (V 0.1.15); (4) a window of 50 SNPs, a step size of ten SNPs, and an $r^2$ threshold of 0.2 were used to prune the SNPs using Plink (v1.9). As a result, a set of 415 accessions and 30,282 high-quality SNPs were retained and used for subsequent analyses (Table S2). We connected all the scaffolds into 20 pseudo-groups (Table S7).

## Phenotypic data collection

The 415 accessions were measured for leaf size for association mapping. We measured the ten representative mature leaves length (MLL) and mature leaves width (MLW) of each individual and calculated their means to represent the traits of MLL and MLW, respectively. Leaf size measurements were performed in Spring (monthly average temperature is 14.68 °C) and Fall (monthly average temperature is 21.29 °C), 2018, seperately. Leaf shape index (MLSI), calculated as MLL/MLW, was the third trait we measured. The fourth trait, mature leaf area (MLA), was calculated as $0.75 \times$ MLW $\times$ MLL. We modified our data collection method according to guidelines of UPOV (International Union for the Protection of New Varieties of Plants 2008) for tea. Statistical analysis of the phenotypic data was conducted using Microsoft Excel 2010 and SPSS 15.0 software. Pearson's correlation coefficients (r) were calculated for the four traits. Means, standard deviations, and ranges were calculated for trait distribution. To evaluate whether the data followed a normal distribution, Skewness and Kurtosis were calculated using the Descriptive Statistics model-based Frequencies distribution of analysis implemented in SPSS15.0 software.

## Genetic purity, distance, and relationships of accessions

The following equation was used to calculate the polymorphism information content (PIC) values for the SNP data (Bostein et al., 1980). $Pi$ and $Pj$ were the frequencies of the number of $i$ and $j$ alleles respectively, and n was the number of alleles (*Botstein et al., 2008*). We can evaluate each marker locus for its PIC by summing the mating frequencies multiplied by the probability that an offspring will be informative. Under our assumptions, the expected value of PIC can be calculated as

$$\text{PIC} = 1 - \sum_{i=1}^{n} P_i^2 - \sum_{i=1}^{n-1} \sum_{j=i+1}^{n} 2P_i^2 P_j^2$$

The alleles, genetic distance and the observed heterozygosity (Ho) were calculated on every groups with TASSEL v.5.2.37 (*Bradbury et al., 2007*). The neighbor-joining cluster analysis were performed using DARwin v.6.0.17.

## Development of core collections

As subsets of larger genetic collections, core collections contain the smallest number of accessions that represent the maximum diversity of the raw collection. DARwin (v.6.0. 17) was used to construct the diversity trees (*Hamon et al., 2003*). We used 10,000 bootstraps to determine dissimilarities and transformed them into Euclidean distances. The phylogenetic tree was constructed based on the 30,282 SNPs using the unweighted Neighbor Joining (NJ) method. Then, the 'maximum length subtree function' was used to generate the core collection as described previously for cowpea (*Egbadzor et al., 2014*), prunus (*Campoy et al., 2016*), sorghum (*Claire et al., 2013*) and rice (*Ndjiondjop et al., 2017*). The maximum length subtree implemented a stepwise procedure that consecutively prunes redundant units. In this procedure, sample size determination, which maintains the largest diversity, is allowed, and the procedure is visualized in the phylogenetic tree of the original population of all 415 accessions. If the distance between two accessions, as judged by the edge length, is

small, they are considered redundant. Because of more uncommon characters, accessions with the longest edge are considered the most diverse. A 'removed edge value' offered by the NJ tree was used to identify presumptive clusters of synonym accessions, and a threshold value of 0.0008 (Euclidean distance) was required to confirm the synonym mentioned. We determined the final core set that represents the maximum genetic diversity based on the pruned edge length of the initial tree length and the sphericity index (*Hamon et al., 2003*).

## Association analysis

Association analysis was carried out on the four measurable traits, namely MLL, MLW, MLSI, and MLA, in two seasons, independently. We used the mixed linear model (MLM) implemented in TASSEL (v.5.2.37) because the GLM model has a high false positive rate (*Bradbury et al., 2007*) following the user manual. To overcome this limitation, a comparison was drawn between the two association analysis models using TASSEL 5.2.43, and the most optimized model for each analyzed quality trait was identified and used for subsequence analyses (*Bradbury et al., 2007*; *Heena et al., 2018*). To control the potential false-positives result from the confounding of population structure, PCA-matrix or Q-matrix was used as the fixed effect in MLMs. Kinship matrix (K) was considered as a covariate factor in MLMs to cut down the rate of false positives among genotypes (Yu et al., 2006). If the significance threshold of a SNP has the lowest $P$-value in the peak area ($P \leq 0.05/30,282 = 1.65 \times 10^{-6}$), it is considered significantly associated with the targeted trait. Correlation coefficient ($R^2$) explained the phenotypic variation by every marker-trait association analyzed.

## RESULTS

### Genetic purity, distance, and relationships

GBS analysis of the 415 tea accessions was conducted using IlluminaHiSeq X Ten and the relevant information was described in our previous study (*Niu et al., 2019*). Observed heterozygosity in each sample ranged from 0.041 to 0.386 (Fig. S2), with an average heterozygosity of 0.225. Among the 415 accessions, 73.5% had an observed heterozygosity >0.201 (Table S1). The pairwise genetic distance between two accessions ranged from 0.088 to 0.298 (Table S1), with an average genetic distance of 0.249 among all accessions (Fig. 1; Table S1). The genetic distances of 36.6% accession pairs were between 0.200 and 0.250, and those of 3.73% and 59.68% accession pairs were <0.200 and >0.250, respectively (Fig. 1; Table S2). Based on the NJ cluster analysis on the genetic distance matrix, the 415 accessions could be clustered into fourteen groups (Fig. 2A; Table S1) based on cultivation status (ancient landraces, wild tree, modern landraces and breeding cultivars), growth habits (cultivation type and wild type), and species, and the here-derived core and mini-core collections of origin. Group one consisted of 71 accessions, most of which were cultivation type (of the group one, 85.92% are cultivation type), ancient landraces (of the group one, 74.65% are ancient landraces), and *C. sinensis* (of the group one, 85.92% are *C. sinensis*). The second group contained 88 accessions, most of which belong to *C. tachangensis* (of second group, 93.18% are *C. tachangensis*) and wild type (of the second group, 96.59% are

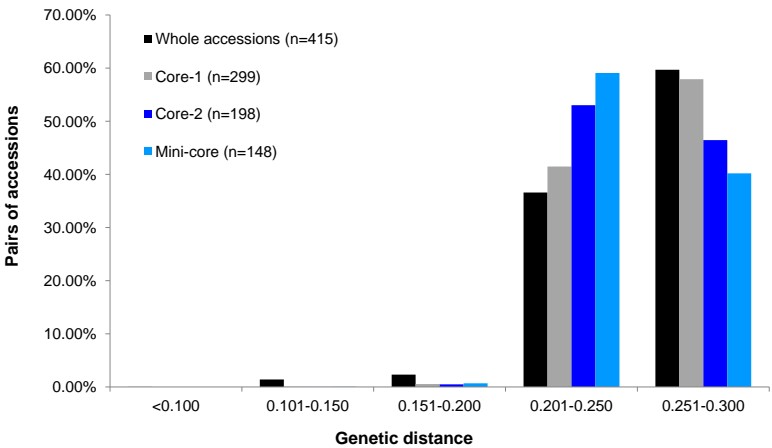

**Figure 1** **Frequency distribution categories of pairwise genetic distance of 415 tea accessions.** Frequency distribution categories of pairwise genetic distance of 415 tea accessions (black fill); a core set of 299 accessions (grey fill); a core set of 198 accessions (blue fill), and a mini-core set of 148 accessions (light blue fill), based on 30,282 polymorphic SNPs.

wild type). The third group consisted of 54 accessions, most of which were wild type (of the third group, 98.15% are wild type) and *C. remotiserrata* (of the third group, 91.48% are *C. remotiserrata*). Group four had a total of 76 accessions and most were cultivation type (of the group four, 97.37% are cultivation type), modern landraces (of the group four, 61.84% are cultivation type) and *C. sinensis* (of the group four, 98.68% are *C. sinensis*). Group five had 54 accessions, all of which were from *C. sinensis* and cultivation type, and the number of modern landraces were almost the same as that of the ancient landraces. Groups six, seven, eight, 12, 13 and 14 consisted of 15, 15, two, seven, one and one accessions, respectively, all of which were ancient landraces, cultivation type, and *C. sinensis* varieties. Groups nine and ten consisted of three and six accessions, respectively, which were wild tree and *C. remotiserrata.* Group 11 had a total of 20 accessions and most were cultivation type (of the Group 11, 95.45% are cultivation type), ancient landraces (of the Group 11, 95.45% are ancient landraces) and *C. sinensis* (of the group 11, 90.91% are *C. sinensis* varieties) (Fig. 2A; Table S1).

## Creating core and mini-core collections

We proposed genetic core and mini-core sets to represent the genetic diversity of a large tea population. These core and mini-core sets can be used for association studies, breeding, and other purposes (*Brown, 1989*). The maximum length subtree method implicated in DARwin v.6.0.17. was repeatedly applied to remove the most redundant accessions until the sphericity index percentage and pruned edge were linear with a relatively low slope, corresponding to 299 accessions (Fig. S4). The 299 core accessions could represent the 415 accessions (referred to as 'core-1' from hereafter) (Table S1 ; Fig. 2B). Then, the sphericity index increases stably and slowly from 301 to 198 on the *x*-axis, indicating no significant difference in the information of the 104 accessions, suggesting that the elimination of these accessions had no significant impact on sphericity index (Fig. S4). The 198 core accessions
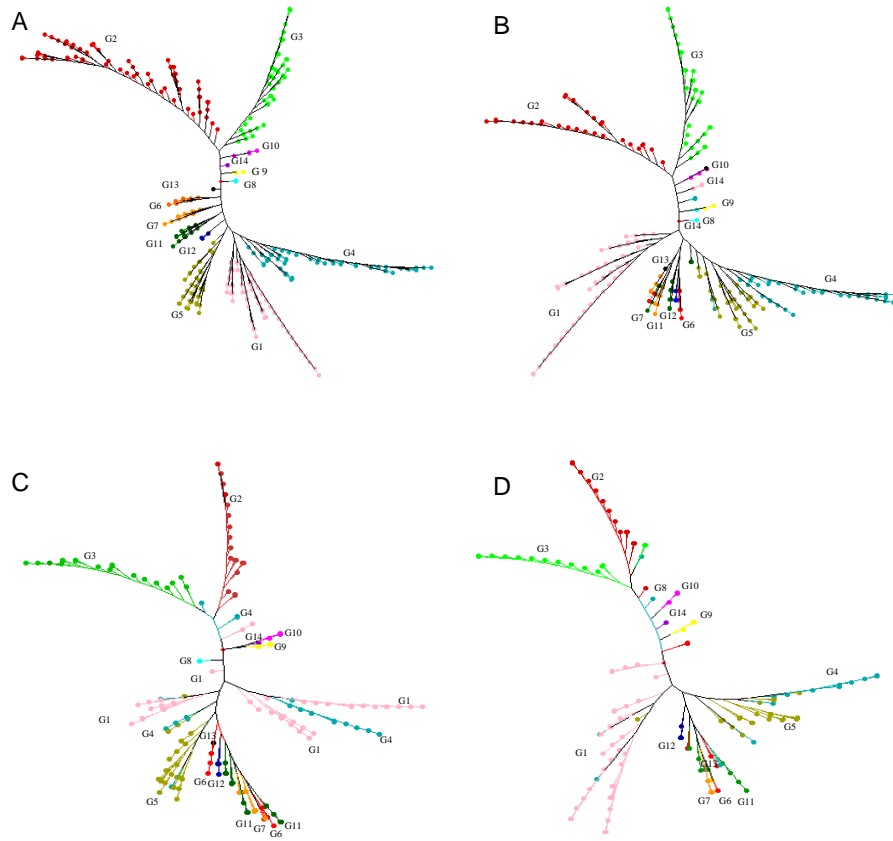

**Figure 2** **Neighbor joining tree.** Neighbor joining tree for (A) 415 accessions (G1 = Group 1, pink; G2 = Group 2, red;G3 = Group 3, light green; G4 = Group 4, blue; G5 = Group 5, brown ; G6 = Group 6 , orange red; G7 = Group 7 , orange yellow; G8 = Group 8, light blue; G9 = Group 9 ,yellow; G10 = Group 10, light purple; G11 = Group 11, dark green; G12 = Group12, dark blue; G13 = Group 13, black; G14 = Group 14, purple), (B) 299 accessions selected for a core set, (C) 198 accessions selected for a core set and (D) 148 accessions selected for a mini-core set based on 30,282 polymorphic SNPs.

(referred to as 'core-2' hereafter, which contains less accessions than 'core-1') were selected to represent the 415 accessions (Fig. 2C; Table S1). The percentage of sphericity index and pruned edge flattened until reaching a sample size of 146 accessions on the *x*-axis (Fig. S4). Thus, the 146 remaining accessions constitute the mini-core set. We suggest that these 148 accessions, consisting of the 146 retained accessions and two cultivars ('Fudingdabaicha' and 'Tieguanyincha'), represent the mini-core set of the Guizhou Plateau tea germplasm (Fig. 2D; Table S1). The selected cores and mini-core sets originated from 14 different clusters yielded from NJ tree, respectively, although some individuals were assigned to other clusters (Fig. 2), which suggested that edges constituting the phylogenetic tree backbone have been retained.

## PeerJ

**Table 1** Genetic differentiation of core and mini-core sets of tea plant in Guizhou Province.

| Group | S | ASC | Ho | PIC | MAF | GDR | AGD |
|---|---|---|---|---|---|---|---|
| Whole Set | 415 | 26,810 | 0.215 ± 0.006a | 0.359 ± 0.006a | 0.171 ± 0.002a | 0.088–0.298 | 0.249 |
| Core-1 | 299 | 26,592 | 0.223 ± 0.005a | 0.360 ± 0.008a | 0.172 ± 0.001a | 0.110–0.298 | 0.252 |
| Core-2 | 198 | 24,917 | 0.206 ± 0.006a | 0.363 ± 0.006a | 0.170 ± 0.001ab | 0.115–0.298 | 0.250 |
| Mini-Core | 148 | 23,731 | 0.193 ± 0.006b | 0.355 ± 0.005a | 0.168 ± 0.001b | 0.115–0.292 | 0.248 |

**Notes.**

S, Sample size; ASC, Average Site Count; Ho, observed heterozygosity; PIC, polymorphism information content; MAF, Minor Allele Frequency; GDR, Genetic distance range; AGD, average genetic distance.

The different letters indicate a significant difference in a column at $p = 0.05$ levels by $T$-test.

## Diversity assessment of the cores and mini-core sets

The alleles and genotypes, PIC, genetic distances and minor allele frequency of the 415 accessions were compared to those of the two core sets and the mini-core set. With regard to the average polymorphic sites in the 415 tea accessions, the reduction in sample size from 415 to 299 and 198 accessions in the two core sets, and 148 in the mini-core set reduced SNPs numbers by only 103 (0.36%), 1,403 (4.93%), and 1,722 (6.04%), respectively (Table 1). Ho, PIC and MAF in Core-1 and Core-2 sets were almost the same as for whole tea accessions, suggesting that both core-1 and core-2 can represent the 415 accessions. Ho, PIC and MAF in mini-Core set were 89.77%, 98.89% and 98.25% of those of the whole population, respectively (Table 1). Allele and genotype frequencies in the two core sets and one mini-core sets almost made no odds from the entire set (Fig. S3). Consistent with our hypothesis, the genetic distances between accession pairs increased slightly in the core and mini-core sets (Fig. 1; Table S2–S5). The proportion of accession pairs with a genetic distance greater than 0.15 was 98.59% for the entire population, compared to 99.95%, 99.97% and 99.95% in core-1, core-2 and the mini-core set (Fig. 1; Table 1), respectively. The proportion of samples number removed per group based on cluster analysis of NJ tree was not consistent in the two core and one mini-core sets (Fig. 2; Table 2; Table S1). The proportion of accessions removed from group two and group four were significantly higher than that from groups 1, 3, 5, 6, 7, 10, 11 and 12. The numbers of accessions excluded from groups 8, 9, 13 and 14 in the two core sets and the mini-core set were same as that from the entire tea collection (Fig. 3). The mini-core collection consisted of the accessions from all 14 groups, as revealed by the cluster analysis of NJ tree, and represented 14 groups, three cultivation status, two growth habits, and four different species (Fig. 3). These results indicated that the core and mini-core collections can well represent the 415 accessions.

## Genome-wide association studies

Skewness and Kurtosis for MLL, MLW and MLSI did not differed from those of a normal distribution (Table 3). Despite the existence of left obliqueness, MLA still exhibited normal distribution. (Table 3). The four traits were significant correlated ($P < 0.0001$), as revealed by the Pearson correlation analysis (Table 4). MLSI was highly correlated with MLL and MLW but not with MLA. The same trait was significant correlated ($P < 0.0001$) between two environments (Table 4).

**Table 2** The number of whole accessions and core collections distributed in 14 groups.

| Group | Number of accessions of whole set | Number of accessions retained in the 299 core set | Number of accessions retained in the 198 core set | Number of accessions retained in the 148 mini-core set |
|---|---|---|---|---|
| Group 1 | 71 | 64 | 49 | 37 |
| Group 2 | 88 | 38 | 19 | 16 |
| Group 3 | 55 | 40 | 22 | 15 |
| Group 4 | 75 | 53 | 24 | 20 |
| Group 5 | 54 | 44 | 33 | 26 |
| Group 6 | 15 | 14 | 12 | 5 |
| Group 7 | 15 | 11 | 8 | 6 |
| Group 8 | 2 | 2 | 2 | 2 |
| Group 9 | 3 | 3 | 3 | 3 |
| Group 10 | 6 | 3 | 3 | 2 |
| Group 11 | 22 | 19 | 16 | 11 |
| Group 12 | 7 | 6 | 5 | 3 |
| Group 13 | 1 | 1 | 1 | 1 |
| Group 14 | 1 | 1 | 1 | 1 |
| Total | 415 | 299 | 198 | 148 |

The population structure of the 415 accessions was studied with STRUCTURE 2.3.4 on the basis of the LD-pruned 1,135 high-quality SNPs. Three subpopulations were acquired, which was coincident with the preceding reports (*Niu et al., 2019*). As a result, the Q-matrix was generated from $k = 3$ using STRUCTURE 2.3.4 and used for GWAS as the fixed effect. P-matrix, as the first three PCs value given by the PCA analysis, was used as covariates in the GWAS model. We also calculated kinship matrix using TASSEL. Therefore, the MLMQ+K and MLMP+K models were compared based on the Q-Q plot (Fig. S5) and the MLMP+K model best fitted the four traits.

We identified nine SNPs that significantly associated with MLL, MLW, MLA and MLSI ($P < 1.655E{-}06$) (Fig. S6 ; Table 5; Table S6). Among these, three were significantly associated with multiple traits. In the case of MLL, significant associations were detected for SNPs PG_5:79474508 (on xfSc0000122) and PG_1:32346865 (on Sc0000000), which explained 3.77% and 3.23% of the phenotypic variation, respectively, and had a deduced effect of major alleles from −0.72 to −2.94 (Table 5; Table S6).

For MLW, only one significant SNP was detected on the same locus as MLL, which explained 8.24% of the phenotypic variation; this SNP has a negative major allele effect of −1.31 and a positive minor allele effect of 0.44 for MLW. Six loci, located at Sc0000000, Sc0000365, Sc0002172, Sc0002452, Sc0004093 and xfSc0000122, significantly correlated with MLA, explaining 3.49% to 4.50% of the total phenotypic variation, with the main influence varying from -18.23% to 4.462% and minor effects varying from -18.76% to 42.47% (Table 5; Table S6). Three SNPs located at Sc0000037, Sc0000011 and Sc0000046 were found to significantly associate with MLSI, explaining 4.012%, 4.33%, and 3.83% of the MLSI variation at 82,813,415, 30,871,283, and 98,891,957 bp, respectively; while the deduced effect of the major alleles ranged from -0.08 to 0.06 (Table 5; Table S6). Among

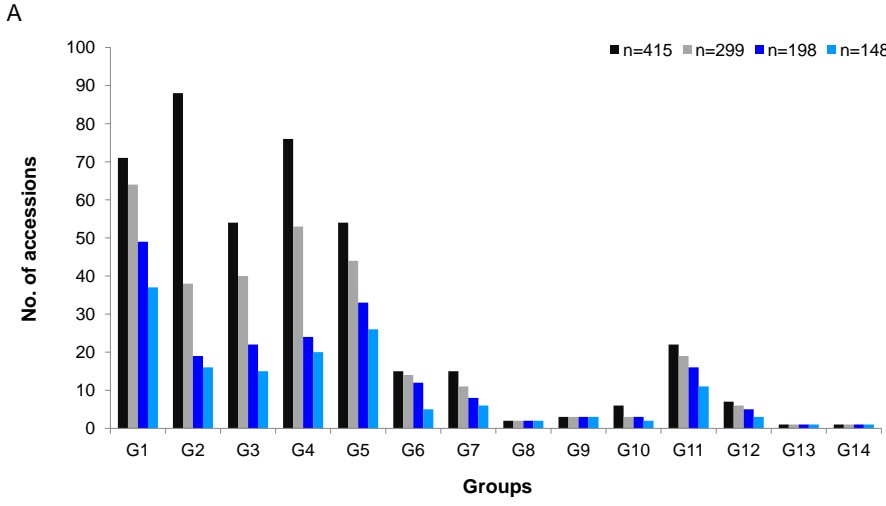

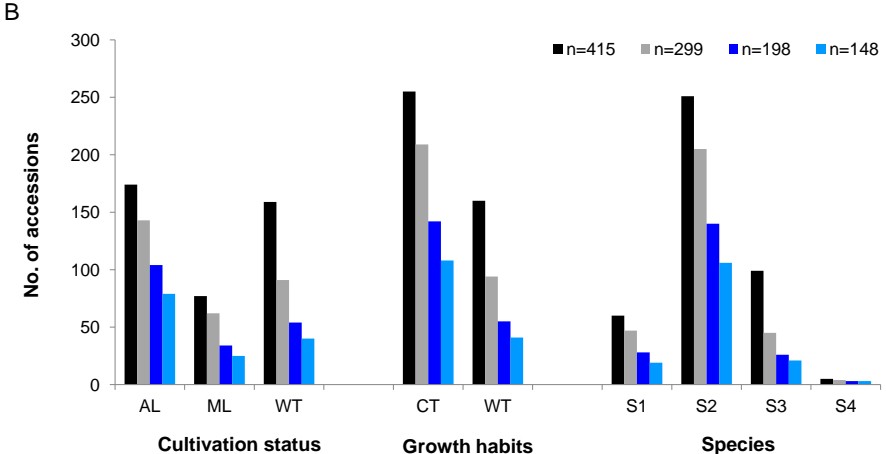

**Figure 3** **Summary of accessions elected to conform core-1, core-2 and mini-core.** Summary of accessions elected to conform core-1, core-2 and mini-core collection compared with the whole tea accessions by (A) groups predicted based on cluster analysis (G1 = Group 1, G2 = Group 2, G3 = Group 3, G4 = Group 4, G5 = Group 5, G6 = Group 6 , G7 = Group 7 , G8 = Group 8, G9 = Group 9 , G10 = Group 10, G11 = Group 11, G12 = Group 12, G13 = Group 13, G14 = Group 14) ; (B) summary of accessions selected to account for core-1, core-2 and mini-core collection compared with the whole tea accessions in each group by cultivation status (AL, Ancient landraces; ML, Modern landraces; WT, Wild tree ) , by Growth Habits (CT, cultivation type; WT, wild type) and Species of origin (S1, *C.sinensis*; S2, *C.tachangensis*; S3, *C.remotiserrata*; S4, *C.taliensis*).

these SNPs, two were co-associated with multiple traits. PG_5:79474508, which had a negative effect of major alleles, was co-associated with MLL and MLA. PG_1:32346865 with an increased effect of major alleles was co-associated with MLL, MLW and MLA (Table 5; Table S6). Three, three and two significantly associated SNPs identified in the whole population were retained in core-1, core-2 and mini-core set, separately (Table S7).

**Table 3** Trait statistics collected for mature leaf length (MLL), mature leaf width (MLW), mature leaf shape index (MLSI) and mature leaf area (MLA).

| Trait | Mean | Std.dev | Minimum | Maximum | Skewness (Std.Err) | Kurtosis (Std.Err) |
|-------|------|---------|---------|---------|--------------------|--------------------|
| MLL | 8.655 | 2.251 | 3.886 | 15.200 | 0.497(0.122) | 0.002(0.243) |
| MLW | 3.748 | 0.933 | 1.986 | 6.830 | 0.687(0.122) | 0.415(0.244) |
| MLA | 25.700 | 12.929 | 6.570 | 75.566 | 1.223(0.122) | 1.582(0.244) |
| MLI | 2.326 | 0.316 | 0.830 | 3.457 | 0.043(0.122) | 1.949(0.244) |

**Table 4** The correlation coefficient between two environment and among four traits.

| Trait/ Environment | MLL | MLW | MLA | MlSI |
|--------------------|-----|-----|-----|------|
| MLL | 0.896[**] | – | – | – |
| MLW | 0.857[**] | 0.960[**] | – | – |
| MLA | 0.953[**] | 0.951[**] | 0.905[**] | – |
| MLSI | 0.338[**] | −0.174[**] | 0.081 | 0.806[**] |

**Notes.**

Note: The diagonal line is the correlation coefficient of two environments

[*]Indicate a significant difference in a column at $p = 0.05$ levels.

[**]Indicate a significant difference in a column at $p = 0.01$ levels.

## DISCUSSIONS

### Genetic purity, genetic distance and relationships

GBS is a cost-efficient and effective genotyping method (*Yang et al., 2017*; *Bhattarai & Subudhi, 2018*; *Hackett et al., 2018*). In this study, 390.30 Gb clean reads were generated. Among them, 0.208–3.32 Gb clean reads were generated after the quality filtering step in 415 accessions, and 30,282 high-quality SNPs were retained with strict filter conditions. More SNPs were identified in this study compared with previous studies, which meets the requirements of our objectives (*Chen et al., 2017*; *Eltaher et al., 2018*).

*Camellia sinensis* and related species are cross-pollinated species with a low self-pollination rate ranging from 2% to 6% (*Chen, Yu & Tong, 2000*; *Ma et al., 2018a*; *Ma et al., 2018b*; *Ma et al., 2018c*). Consistently, the heterozygosity rate of 73.5% were >20%. Most alterations in allele frequencies occurred during natural regeneration without human intervention, which is in agreement with previous studies (*Niu, 2014*), indicating that the tea population from the center of origin maintained a high level of genetic diversity. Genetic distance measures the genetic divergence between a given pair of accessions or populations; the pairs usually share many alleles with a small genetic distance (*Ndjiondjop et al., 2017*). The results in this study suggested remarkably different genetic distances between pairs of 415 accessions, with 3.73% of the pairs in similarity, 36.6% of the pairs moderately distant, and 59.68% significantly distant. We detected very low redundancy in our collection, which consists with the low observed genetic divergence and high genetic variation retained from the original tea population. Using clustering, we identified 14 groups or clusters to establish the core accessions. In general, the groups clustered according to the cultivation status, growth habits, and species.

Niu et al. (2020), *PeerJ*, DOI 10.7717/peerj.8572

**Table 5** SNPs significantly associated with mature leaf length (MLL), mature leaf width (MLW), mature leaf shape index (MLSI) and mature leaf area (MLA) detected by GWAS of 415 accessions.

| Trait | SNP Marker | Pse-Group | Position (bp) | Scaffold | Major/minor allele | MAF | *P*-value | Major/minor allele effect | *R*-square | No. of accessions with homozygous major/minor allele | No. of accessions with heterozygous allele |
|---|---|---|---|---|---|---|---|---|---|---|---|
| MLL | PG_13:79474508 | 13 | 79474508 | xfSc0000122 | G/T | 0.07 | 5.41E−07 | −0.72/−3.36 | 3.77% | 307/14 | 22 |
| | PG_1:32346865 | 1 | 32346865 | Sc0000000 | T/A | 0.05 | 1.03E−06 | −2.94/1.09 | 3.23% | 345/8 | 23 |
| MLW | PG_1:32346865 | 1 | 32346865 | Sc0000000 | T/A | 0.05 | 3.59E−07 | −1.31/0.44 | 8.24% | 344/8 | 23 |
| MLA | PG_1:32346865 | 1 | 32346865 | Sc0000000 | T/A | 0.05 | 1.95E−08 | −18.23/8.25 | 4.50% | 344/8 | 23 |
| | PG_4:40545730 | 4 | 40545730 | Sc0000365 | G/A | 0.07 | 8.80E−07 | −1.47/14.72 | 3.79% | 316/13 | 20 |
| | PG_12:21050315 | 12 | 21050315 | Sc0002172 | C/T | 0.06 | 9.47E−07 | −4.12/11.61 | 3.58% | 327/19 | 8 |
| | PG_12:133979444 | 12 | 133979444 | Sc0002452 | G/A | 0.06 | 1.07E−06 | 1.06/42.47 | 3.49% | 322/2 | 36 |
| | PG_16:74477057 | 16 | 74477057 | Sc0004093 | G/A | 0.08 | 1.40E−06 | 4.46/31.33 | 3.81% | 283/4 | 42 |
| | PG_13:79474508 | 13 | 79474508 | xfSc0000122 | G/T | 0.07 | 1.45E−06 | −5.57/-18.76 | 3.68% | 306/14 | 22 |
| MLI | PG_13:82813415 | 1 | 82813415 | Sc0000037 | T/C | 0.13 | 3.24E−07 | −0.08/-1.25 | 4.01% | 266/1 | 96 |
| | PG_17:30871283 | 17 | 30871283 | Sc0000011 | A/G | 0.05 | 5.84E−07 | −1.49/0.09 | 4.33% | 280/1 | 32 |
| | PG_10:98891957 | 10 | 98891957 | Sc0000046 | G/A | 0.06 | 1.40E−06 | 0.06/-1.50 | 3.83% | 306/1 | 36 |

## Selection of core and mini-core

Establishing the core and the mini-core sets with the lowest level of redundancy that represent the maximum potential genetic diversity from the total collection facilitates the identification of suitable variations for GWAS and MAB (*Brown, 1989*; *Ndjiondjop et al., 2017*). Core collections could be evaluated based on genetic markers or phenotypic traits, including pairwise distances (*Hamon et al., 2003*; *Franco et al., 2005*; Leroy et al., 2014) and allelic richness/diversity (*Beukelaer et al., 2012*; *Beukelaer, Davenport & Veetle, 2018*). Selection of the most suitable evaluation method depends upon the purpose of core collections (*Odong et al., 2013*). Preserving most alleles is an ideal way to conserve germplasm, while the approaches based on distance majorly hammer the retention of most combinations of alleles in specific genotypes, which are suitable for GWAS and MAB (*Campoy et al., 2016*; *Ndjiondjop et al., 2017*). Thus, we used the distance-based methods applied on GBS data to propose the first core collection and the mini-core collection from the tea origin center, the Guizhou plateau, and accounting for wild trees, ancient landraces, modern landraces and cultivars in this study. Core-1 and Core-2 collections and one mini-core collection were created based on Sphericity index and the length of pruned values, which contain 72%, 48% and 36% of the total number of accessions. Ho, PIC, MAF, allele and genotype frequencies in Core-1 and Core-2 sets were almost the same as for the whole population, which suggested that both core-1 and core-2 can well represent the 415 accessions for further study. The genetic diversity parameters, genetic distances , allele information and retained SNPs of core-1 and core-2 revealed that the core-2 of 198 accessions was equally appropriate for representing the whole population as core-1 of 299 accessions Therefore, the core-2 of 198 accessions was chosen as the appropriate set, considering the costs of future research. The mini-core set selected in this study is easy to manage for phenotypic and physiological evaluation in the field and under controlled conditions when selecting parent lines for improving the traits of interest and identifying genes associated with these traits using GWAS.

## Genome-wide association studies

Based on morphology, tea plants can be divided into two subgroups. Small-leaf shrubs that are cold tolerant constitute group one, whereas group two consists of large-leaf arbor trees that are less resistant to cold (*Yao et al., 2012*). Leaf size is associated to the fitness and temperature-adaptation history of tea varieties (*Tan et al., 2016*). Leaf size may also relate to tea production and the shape of dry tea. The mapping population used in this study included two sub groups based on leaf size. Therefore, we observed a high level of segregation in all four leaves size-related traits. Quantitative leaf traits are influenced by the environment (*Tan et al., 2016*; *Baker et al., 2015*). In this study, the same trait was significantly correlated between two different environments, suggesting that there are little false positive in the significantly associated SNPs.

MLM was employed for association mapping and two models (MLMQ+K and MLMP+K) to control false positive caused by population structure. Leaf size is associated with local adaptation, and it can actually decrease the false rate by removing accessions with extreme genetic diversity and phenotypic expression (*Pace et al., 2015*). We did not

test the GLM model but used the combination of MLMQ+K and MLMP+K models, which reduce the rate of false positives for detecting SNPs, and are proven to be more successful than using each strategy alone (*Song et al., 2019*). Q denotes the Q-matrix produced from the population structure ($k = 3$) and P represents P-matrix from the top three PCs, that are used as covariates in GWAS. K refers to kinship matrix and is used to determine the correlations between individuals. Both the $Q+K$ and $P+K$ matrices fit the MLM to control spurious associations resulted from relatedness and population structure, separately (*Zhao et al., 2019*).

In previous studies, tea germplasm showed significant variations for leaf size (*Yao et al., 2012*). However, to our knowledge, few genetic loci influencing this trait have been identified. *Tan et al. (2016)* reported one major QTL (*qLSI13*) for leaf size in tea plant. In this study, nine SNPs significantly associated with leaf traits were identified using the P+K MLM. Six, three, two and one SNPs were significantly associated with MAL, MLSI, MLL and MLW, respectively. SNP PG_1:32346865 was significantly associated with three traits, MLL, MLW and MLA, all were significantly positively correlated ($P < 0.0001$). Which was consistent with the results of significant positive correlation among the three characters (Table 4). Three SNPs significantly associated with MLSI were carrying with only 1 minor allele, which need to keep the work in exploring the truth in next step study.

Our study provides evidence that leaf size is affected by major effect genes, which was consistent with previous reports (*Tan et al., 2016*). These results lay a foundation for future study and will inform leaf size-related gene mining and MAB in tea breeding.

## CONCLUSIONS

This study illustrates that GBS is efficient for analyzing the genetic purity, distance and relationship of tea germplasm, and for creating the core sets. The core and mini-core collections account for approximately 47% and 37% of the entire collection, respectively, and contains 93–95% of all SNPs and almost all genotype frequencies and alleles that were observed in the entire tea collection. These core sets are highly valuable for identifying favorable alleles and selecting parent lines to improve agronomically important traits in tea varieties. Our study identified SNP markers associated with leaf size. These SNPs can be employed in MAB for tea improvement.

## ACKNOWLEDGEMENTS

We thank tea office of Guiding, Huishui, Liping, Renhuai, Sandu, Wuchuan, hishui, Daozhen, Dejiang,Duyun, Guian, Jinsha, Liuzhi, Nayong, Pu an, Puding ,Qinglong, Qixingguan, Sandu, Shiqian, Shuicheng, Tongzi, Wuchuan, Xingren, Xingyi, Xishui, Yanhe, Yinjiang, Yuqing, Zhenfeng, Zheng'an for their help in teas collection. We thank the College of Tea Science of Guizhou University and the Department of Horticultural Sciences of Texas A&M University for providing research facilities and computing facilities.

### Funding

This work was supported by the National Science Foundation of China (31560222), the Science and Technology Plan Project of Guizhou province (20172558, 20172557, 20175788, 20191404), the USDA-NIFA SCRI grant (2017-51181-26834), the Guizhou Top Level Innovation Talents Cultivation Project (20164003), the Talent Base for Germplasm Resources Utilization and Innovation of Characteristic Plant in Guizhou (RCJD2018-14) and the Genetically Breeding Major Project of the Ministry of Agriculture of China (2016ZX08010-003). The funders had no role in study design, data collection and analysis, decision to publish, or preparation of the manuscript.

### Grant Disclosures

The following grant information was disclosed by the authors:
National Science Foundation of China: 31560222.
Science and Technology Plan Project of Guizhou province: 20172558, 20172557, 20175788, 20191404.
USDA-NIFA SCRI grant: 2017-51181-26834.
Guizhou Top Level Innovation Talents Cultivation Project: 20164003.
Talent Base for Germplasm Resources Utilization and Innovation of Characteristic Plant in Guizhou: RCJD2018-14.
Genetically Breeding Major Project of the Ministry of Agriculture of China: 2016ZX08010-003.

### Competing Interests

The authors declare there are no competing interests.

### Author Contributions

- Suzhen Niu conceived and designed the experiments, performed the experiments, analyzed the data, prepared figures and/or tables, authored or reviewed drafts of the paper, and approved the final draft.
- Hisashi Koiwa, Degang Zhao and Zhengwu Chen conceived and designed the experiments, authored or reviewed drafts of the paper, and approved the final draft.
- Qinfei Song and Ying Wang performed the experiments, analyzed the data, prepared figures and/or tables, and approved the final draft.
- Dahe Qiao and Juan Chen performed the experiments, prepared figures and/or tables, and approved the final draft.
- Tianyuan Zhang analyzed the data, prepared figures and/or tables, and approved the final draft.

### Data Availability

The raw sequence data reported in this study are available in the Genome Sequence Archive in BIG Data Center, Beijing Institute of Genomics (BIG), Chinese Academy of Sciences: CRA001438.

## Supplemental Information

Supplemental information for this article can be found online at http://dx.doi.org/10.7717/peerj.8572#supplemental-information.

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
