# Peer review of "Development of core-collections for Guizhou tea genetic resources and GWAS of leaf size using SNP developed by genotyping-by-sequencing"

_PeerJ, doi:10.7717/peerj.8572_

## Round 0.1 · original submission · Major Revisions

We have received the comments from three reviewers, all of them had positive recommendation. However, they also suggested revision is needed. Please take all comments into consideration and make revision accordingly.

Reviewer 1 ·

Basic reporting

1.The title should be more specific and more concise, because the genetic resources are solely from Guizhou province. So, Guizhou province, China must add to the title. The below title is suggested for the authors:
Development of the core collection for Guizhou tea genetic resources and GWAS of leaf size using SNP developed by genotyping-by-sequencing

2. Several important references about tea core collection in China, worldwide are missed.
Wang XC, Chen L, Yang YJ. Establishment of core collection for Chinese tea germplasm based on cultivated region grouping and phenotypic data. Front. Agric. China 2011, 5(3): 344–350
Taniguchi F, Kimura K, Saba T. Worldwide core collections of tea (Camellia sinensis) based on SSR markers. Tree Genetics & Genomes 2014, 10, 1555-1565

Experimental design

no comment

Validity of the findings

3.Line 45-46.
The Guizhou Plateau is the center of origin of tea plants (Wei et al., 2018; Jin et al., 2016). This is not a correct statement, and Jin et al.(2016) did not have this result. So, this sentence should be deleted.

4.Line 103-104,
From the second to the fourth, Camellia should be abbreviated form C.

5.Check all the original references carefully
Line 538-539
Chen, L., Apostolides, Z., and Chen, Z.M. (2013). Global Tea Breeding: Achievements, Challenges and Perspectives. Advanced Topics in Science & Technology in China.

Should be:
Chen, L., Apostolides, Z., and Chen, Z.M. (2012). Global Tea Breeding: Achievements, Challenges and Perspectives. Springer-Zhejiang University Press

Line681
Yao, M.-Z., Ma, C.-L., Qiao, T.-T., Jin, J.-Q., and Chen, L. (2011) should be (2012)

Additional comments

The construction of core collection for Guizhou tea genetic resources is very important for the conservation and utilization of tea germplasm.
Anyway, the title should be re-written to be more specific and concise. The references should also to be checked carefully.

Annotated reviews are not available for download in order to protect the identity of reviewers who chose to remain anonymous.

Reviewer 2 ·

Basic reporting

This manuscript describes a population level study using collected tea plants. Authors displayed the diversity of their collected samples to some extent, and a diverse collection of samples will be useful for the community. The study is potentially meaningful, however, many accession are not pure enough. The data generated in this study will be difficult to be reused and should be discussed or mentioned. I have several suggestions for this manuscript.

1. Please considering to shorten the title to be more concise.
2. Many places of English/scientific writing are weird and need to be improved, for example, L44, “Tea has,” there is no need to add comma after has; L59, “Compared with other means”, means is a weird word used in here; L60, “Finding yielded”? I can not understand this; L101, “five” should be replaced by 5. Places like these need to be caught carefully and fixed.
3. Figure 1, other than labelled the number for each color, please also label the name of your defined collection.
4. For figure 3, it is less informative to show this as a main figure, since all collections for any of allele or allele ratio are almost the same.
5. For table 5, please mention clearly what the highlighted red text means?
6. What is the SNP naming rule in your analysis? If I find one SNP such as PG_1:32346865, how could I refer it to the genotyping data you provide?
7. L290, you mentioned Niu et al.2019 as one of citations, but I can not find it anywhere.
8. L322-L323, please give a citation for the low self-pollination rate.

Experimental design

1. Based on the geographical map provided, I worried there are strongly closed relationship among individuals, please providing a PCA plot with each group labeled.
2. In the supplemental figure, it seems more than half of collected samples have heterozygosity > 0.2, those accessions should not be considered as “accessions”, since they will be hard to propagated. Tea genome is 3.2GB, that means >640MB genome are heterozygous. Many loci will be segregated in progeny. Genotype and phenotype from one of these accessions will could not be maintained consistently through generations.

Validity of the findings

1. I am not able to trace any potential biological meaning for each classified group. Why not use the STRUCTURE suggested subpopulations? If you just list numbers, people will definitely get confused. For some groups, like group 8, 9, 13, 14, there are only a very few accessions, how meaningful they are? Please provide more solid evidence to support this classification.
2. For table 5, you should write clearly how MAF was calculated. Since in this current table, for example, PG_10:98891957, mentioned MAF is 0.06, but 1/306=0.003, which is much lower. How many heterozygous are there? But overall, most of significant SNPs are rare alleles, very weird. Please clearly report how many accessions carrying homozygous majors alleles, how many carrying homozygous minor alleles, and how many carrying heterozygous alleles.

·

Basic reporting

Under the frame of the work entitled „Development of the core collection and genome-wide association analysis (GWAS) of leaf size in tea plant using genome-wide single-nucleotide polymorphism (SNP) developed by genotyping-by-sequencing“ written by Niu et al., tea accessions were mostly collected from the Guizhou region in China. These accessions were phenotypically characterized for breeding relevant traits related to leaf size as well as genotypically profiled by using genotyping-by-sequencing. In addition, core-collections were developed using this data and were subsequently evaluated according to different criteria like, for instance, the amount of retained diversity as compared with the original population, their reduction in size, among other quality criteria for core-collections. Domestication and plant breeding have produced selection bottle necks in most cultivated species, with tea plants not being an exception. For this reason and more than never, studies characterizing genetic resources are highly valuable and could be interesting for a broad audience of readers. The results are in general good presented and further discussed. Unfortunately, I cannot recommend the manuscript for direct publication at its current state, since there are some issues that I found while reading the submitted documents. The most important one refers to the often mentioned statement “These core-collections are suitable for future GWAS studies” which was not directly assessed by the authors in the experimental work. In the following paragraphs I make some suggestions on how to deal with this issue and included additional important minor points. For instance, although the manuscript is relatively well written, there are several sections that need revision. Since the manuscript has no page number, I base the positioning of manuscript sections, that should be assessed by the authors before manuscript acceptance, by using the page numbering of the pdf file that was automatically created by the Journal Managing System of PeerJ.

Most important issue:

The statement “These core-collections are suitable for future GWAS studies” was several times mentioned in the manuscript (see for instance page 18, lines 335-338, or page 19, lines 358-361). In this respect, authors should support this statement by at least reporting the number of marker-trait associations that were identified in the whole population and afterwards retained in core-collections. This will give readers better insights about the power of GWAS using these core-collections.

Minor issues:

Introduction

Page 8, lines 76-79: Please cite the original works of Frankel (1984) and Brown (1989) in this section:
- Brown AHD (1989) Core collections—a practical approach togenetic-resources management. Genome 31(2):818–824.
- Frankel OH (1984) Genetic perspectives of germplasm conservation.In: Arber WK et al (eds) Genetic manipulation: impact on manand society. Cambridge University Press, Cambridge, England,pp 161–170.

Materials and Methods

Page 9, line 110 to page 10, line 117: From what you mentioned in the introduction, tea is a highly heterozygous species, which should be quite similar to that case observed in maze. In maize, this situation leads to a high diversity within accessions, which means that some of them should be interpreted as a sample from a larger populations instead of considering them as defined genotypes. For this reason, it is very typical that several individuals of a same maize accession are characterized and thus represent the diversity pool existent in that particular accession. I know that this is not an easy task and obviously not cheap to assess, but can you at least make a comparison (maybe in the discussion section?) between what is being performed for species like maize and the state-of-the-art in tea? And also mention that this point should be assessed in the future?

Page 11, line 150: After filtering, how did you deal with the remaining missing data? Did you performed all analyses ignoring this issue or did you impute missing values? Please clarify.

Page 11, lines 153-154: What do you mean with 20 groups? Are these linkage groups? Chromosomes? Please clarify

Page 11, line 167: Which kind of measurements did you use to measure skewness and kurtosis? Please specify.

Page 12, line 171: Please explain the equation by describing its different components.

Page 12, line 190: 0.0008 corresponds to an Euclidean distance? Please specify.

Page 13, lines 203-204: I do not understand what do you mean with “the irrelevance among genotypes”. The Kinship matrix accounts for the relatedness between individuals. Please correct this.

Page 13, lines 206-207: How did you calculated the R-squared? Was this base on the mixed model or using a simple linear regression model? Please specify. In addition, R-square is the coefficient of determination, not a correlation coefficient.

Results

Page 13, lines 210-212: I do not think that reporting the size of the datafiles before filtering is so informative. Please mention the average number of read calls per genotype before filtering, or something similar, to give the size of the data a genetic meaning. This also can be applied in the discussion (page 18, line 316).

Page 17, lines 286-287: I do not understand why you wrote “These results suggest that the four traits could be used for association mapping”. Please clarify and justify this idea, otherwise I suggest to eliminate it.

Discussion

Page 19, lines 354-355: “This proportion is significantly higher than those reported in previous studies”. Why do you think this is happening? Could be maybe due to the fact that other populations are too large and have a high level of redundancy, which leads afterwards to a substantial reduction in the number of accessions retained by the core-collection? Please work on this idea and comment in the text.

Page 20, lines 370-371: “Associated traits were characterized under two different environments, suggesting that the significantly associated SNPs were genuine and stable”. I do not agree with this idea. A dataset with two environments is not robust-enough to make such a conclusion. Were the two environments similar or contrasting? How was the correlation of phenotypic values among them? Please clarify and improve.

Page 20, lines 373-375: “Leaf size is associated with local adaptation, and it can actually decrease the false rate by removing accessions with extreme genetic diversity and phenotypic expression”. Please explain in more detail how this decrease in false rate happens.

Page 20, line 376: I do not agree with the idea that the models used offer the resolution that is required for detecting SNPs. Resolution is a property of the population itself and has nothing to do with these models. I suggest to mention here the reduced rate of false positives provided by this model, instead. Please modify.

Page 20, line 381: “assuming an independent relationship between individuals”. This make no sense, please delete.

Page 21, lines 393-394: You already mentioned in page 21 line 386 that Tan et al. (2016) reported one major QTL for leaf size in tea plant. You also found that leaf size is affected by major effect genes, but then say that your findings contradict the ones of Tan et al. (2016). Either you were not clear enough in page 21 line 386, or you are being contradictory. Please be clear and fix this issue.

Page 21, line 397: “This is the first GWAS performed for a tea population”. Please be careful with this kind of statement in the future, because a single search in Google using words “association mapping” and “tea” showed me several studies on GWAS in tea populations. Please modify or eliminate this sentence.

Page 21, line 404: What “linked functional markers” are you talking about? To claim that markers are functional you have to conduct biological validations using complementation or loss-of-function methods. Please modify.

Writing issues or less important issues that could improve legibility of the manuscript if assessed:

Title

Page 5, line 1: I suggest to use “development of core-collections” instead of “development of the core-collection”. Otherwise sounds to me like this is and will be always the one and only tea core-collection developed.

Abstract

Page 6, line 18: Please use “key for” instead of “key to”.

Page 6, lines 19-21: I suggest to present the study in this way “… resources. Genotyping-by-sequencing was used to characterize 415 tea accessions mostly collected from the Guizhou region in China. A total of 30,282 high-quality SNPs were used to estimate the genetic relationships, develop core collections, and perform genome-wide association analysis (GWAS). We suggest 198 and 148 accessions to represent the core set …”

Page 6, line 24: I suggest to use “very well retained” instead of “were almost showed”.

Introduction

Page 7, line 34: Please replace “beverage” with “beverages”

Page 7, line 34: Are tea, coffee and cocoa also the most produced or consumed beverages worldwide? Please add a sentence on this.

Page 7, line 37: Please replace “in family” with “in the family”.

Page 7, line 44: Please replace “Tea has, been” with “Tea has been”.

Page 8, line 60: Please replace “Finding” with “Findings”.

Page 8, line 66: Please check and correct the character size of “the linkage disequilibrium (LD)”, because it is not matching the rest of the text.

Page 8, line 68: Please replace “; other advantages” with “. Other advantages”.

Page 8, lines 69-70: Please replace “and less time-consuming for developing” with “and providing a less time-consuming approach for developing”.

Page 8, line 81: “Criteria basis on” or “criteria based on”? Please check and correct.

Page 8, line 86: I know the reference population used in the study of Muqadassi et al. (2018) very well. In this respect, I can say that this population DOES NOT correspond to a core-collection as defined by Frankel (1984) and Brown (1989). It is actually a diversity panel of European wheat cultivars. I suggest to use another reference in this section or simply eliminate it.

Page 9, line 89: This is the first time in the introduction that you use the term GBS. Please introduce it first a “Genotyping-by-sequencing (GBS)”.

Page 9, line 97: I suggest to stick to either MAS or MAB, but not both, throughout the manuscript. This will reduce the number of abbreviations and terms that you are using.

Materials and Methods

Page 9, lines 105-109: You used the word “referred” four times in less than . I suggest to use some synonyms.

Page 10, line 128: Please replace “of the 12” with “of 12”.

Page 10, line 137: Please replace “basing” with “based”.

Page 11, line 142: Please replace “perl” with “Perl”.

Page 11, line 157: Please move the description in Page 11, lines 161-163 “We measured the length… in March and September, 2018” to the beginning of this paragraph and eliminate the “as described below” phrase. Afterwards you can describe what is MLSI and MLA, otherwise is too confusing.

Page 12, line 174: Please replace “were using” with “were performed using”.

Page 12, line 190: “synonym accessions”. Do you mean “duplicates”?

Page 13, lines 194-195: You already introduced all those abbreviations in the text. Please use only the abbreviations.

Page 13, line 205: Please try to stick to one single format number. For instance, write “30,282” instead of “30282”. Check the format consistency of numbers greater than 1,000 throughout the whole manuscript, figure captions, supplementary material, etc.

Results

Page 13, line 216: Please write “had an observed” instead of “a observed”.

Page 13, line 217-218: Please write “The pairwise genetic distance between two accessions ranged from” instead of “The genetic distance between two accessions within each pair ranged from”.

Page 14, line 224: Please write “the here-derived” instead of “formed”.

Page 14, line 228: Please write “and” instead of “and the”.

Page 14, line 235: Please write “which were” instead of “they were”.

Page 14, line 240: Please write “population. These” instead of “population; these”.

Page 14, line 247: You use the term “variety” interchangeably to make reference to accessions or to cultivars. This could be confusing and lead to misunderstanding. I suggest to use the term “variety” only to make reference about “cultivars”.

Page 15, line 264: Please replace “sets almost were same as the whole” with “set were almost the same as for the whole”.

Page 15, line 266: Please replace “of the whole tea accessions, separately” with “of those of the whole population, respectively”.

Page 15, line 272: Please replace “1). The proportion of samples number removed from per group” with “1), respectively. The proportion of samples number removed per group”.

Page 16, line 283: Please replace “displayed normal distribution” with “did not differed from those of a normal distribution”.

Page 17, lines 297-298: Please replace “For MLL,” with “In the case of MLL, significant association were detected for SNPs” and delete “were detected with these SNPs”.

Page 17, line 301 to page 18, line 302: Please replace “; a negative effect of major alleles of -1.31 and a positive effect of minor alleles of 0.44 were detected” with “. This SNP has a negative major allele effect of -1.31 and a positive minor allele effect of 0.44”. In addition, please specify the trait you are referring to.

Page 18, line 308: Please replace “the deduced” with “while the deduced”.

Discussion

Page 18, line 315: I presume there was problem when the text file was converted into pdf, the word “cost-efficient” has a square instead of a “ff”. Please verify and correct.

Page 18, line 318: Please replace “high-quality SNPs retained” with “high-quality SNPs were retained”.

Page 18, line 321: Please replace “cross-pollination” with “cross-pollinated”.

Page 19, line 331: Please replace “low observed genetic divergence and genetic variation in the tea collection sustained” with “with the low observed genetic divergence and high genetic variation retained from the original tea population”.

Page 19, line 336: Please replace “variety” with “diversity”.

Page 19, lines 345-347: The syntax of the sentence “Compared with … (Odong et al., 2013)“ seems odd and needs to be improved.

Page 19, lines 348-349: Please replace “distance-based methods to propose the first core collection and the mini-core collection from tea origin center, the Guizhou plateau, by GBS, accounting” with “distance-based methods applied on GBS data to propose the first core collection and the mini-core collection from the tea origin center, the Guizhou plateau, and accounting”.

Page 20, line 358: Please replace “appropriate sets considering of the cost of study in the future research” with “appropriate set, considering the costs of future research”

Page 20, line 359: The word “are” is twice in the sentence. Please conserve only one of them.

Page 20, line 365: Please replace “signifies” with “is associated to”.

Page 20, line 369: Please replace “sensitivities to environmental changes” with “influenced by the environment”.

Page 20, line 371: Please replace “genuine and stable” with “environmentally stable”.

Page 20, line 372: Please replace “employ” with “employed”.

Page 20, line 374: Please replace “byremoving” with “by removing”.

Page 21, line 386: Please replace “reported that one” with “reported one”.

Page 21, lines 390-392: I do not understand the text segment “This trend … another (Table 4).”. Please improve the syntax.

Figure and Table legends

General comment: I do not know if this is due to the formatting tool of the Journal, but Figure captions and Table headings appear again several times after page 32. Please double check this issue.

Page 22, lines 412-414: the are several spacing problems with commas and brackets. There must be no space before a comma and there must be a space after the comma. In the case of brackets, there must be a space before an opening bracket “(“ and no space before a closing bracket “)”, whereas there must be no space after an opening bracket and here must be a space after a closing bracket. The only exception is when a closing bracket is followed by a point “.”, because in this case, the closing bracket is directly followed by the point in the way “).”. Please be consistent with this formatting throughout the whole text (manuscript, figure captions, table headings, supplementary files, etc.).

Page 22, line 417: Please replace “entire tea” with “entire collection of tea”.

Page 22, line 417: Please replace “explain” with “conform”.

Page 22, lines 427-428: the term “growth way” is not so clear and can lead to misunderstandings by the readers. I suggest to use instead the term “cultivation status”. Please check this throughout the whole text (manuscript, figure captions, table headings, supplementary files, etc.).

Page 22, line 431: Please replace “plant” with “accessions from”.
Competing interests

Page 24, lines 472-473: “Competing interests” should be in a single line (press enter before the word “Competing”).

Figures and Tables

Page 38, lines 10-11: Please double check that all specific names are written in italics throughout the whole work (manuscript, figure captions, table headings, figures, supplementary files, etc.).

Page 43: In Table 2, the column names are too long and this make the table look odd. Please improve. In addition, in the first column and last row of the table the word “Totle” should be replaced with “Total”.

Page 49: I presume the note “Pre-Group”, Pseudo Group; MAF minor allele frequency” is not complete. Please improve this and include this note as footnote (not in the heading). What is Pse-Group? Pseudo-Group? Again, there are some columns with extremely long names. Please improve.

Experimental design

I included everything in the basic report.

Validity of the findings

I included everything in the basic report.

Additional comments

I included everything in the basic report.

---

## Round 0.2 · Minor Revisions

Some text formatting issues in the manuscript still need to be fixed, in particular Reviewer 3. The reviewers have mentioned this in previous comments, please take this into consideration or your paper cannot progress.

Reviewer 1 ·

Basic reporting

good

Experimental design

good

Validity of the findings

good

Additional comments

This manuscript reported the core collection of tea genetic resources in Guizhou province and the GWAS of leaf size using SNPs by GBS technique. The authors addressed all comment and suggestion raised by the first circule of review. So, it is acceptable for publication in this version for Peer J after a minor revision.
Line 44, "Tea has been cultivated for more than 5000 years in the Yunnan-Guizhou Plateau" should be deleted or changed into “Tea has a long utlization history in the Yunnan-Guizhou Plateau”. Line 45, "Chen et al, 2013" should be “Chen et al. 2012??”
Through the paper, the "variety" should be used correctly, for example “breeding varieties" can be changed into "breeding cultivars", variety can be used with species.

Reviewer 2 ·

Basic reporting

No comment

Experimental design

No comment

Validity of the findings

No comment

Additional comments

Authors addressed most of my concerns. However, for SNPs detected in GWAS result with only 1 or 2 minor allele are really hard to believe they are true. Please drawing violin plot or box plot to show phenotypic difference for individuals carrying these alleles. And authors should also discuss the drawback of these GWAS results in the discussion section.

·

Basic reporting

Please read the attached pdf

Experimental design

Please read the attached pdf

Validity of the findings

Please read the attached pdf

Additional comments

Please read the attached pdf

---

## Round 0.3 · accepted · Accept

After reviewing the comments from reviewers, I recommend acceptance of your manuscript.

Reviewer 1 ·

Basic reporting

good

Experimental design

good

Validity of the findings

good

Additional comments

Line 106, breeding varieties should be changed into breeding cultivars

Reviewer 2 ·

Basic reporting

no comment

Experimental design

no comment

Validity of the findings

no comment

Additional comments

Authors addressed my concerns properly. This current version of manuscript is acceptable to be published.

·

Basic reporting

The authors properly assessed all my most important comments and almost all of my remarks regarding syntax and typos. Although I consider that the manuscript can be already accepted for publication, I kindly would like to ask the authors to take into account the following points before publication:

[Original] General comment: There are still some text formatting issues in the manuscript and you must double check them before sending the new version of your manuscript. There are sections with two blank spaces together (for instance in the reference section).
[A Response] We apologize our carelessness and have checked carefully and corrected point-by-point in text.
[R Response] Please revise again. I still found double blank spaces in the reference section.
[Original] 21. Page 9, lines 221 to 222: it should be “to cut down the rate of false positives” instead of “to cut down the false positive among genotypes”.
[A Response] We have revised in the text (Materials and Methods section, line214, page9).
[R Response] Please revise again, the mistake is still there. You cut down the rate of false positive associations, not of false positive genotypes. Or if you prefer it, you can say “to correct for relatedness among genotypes”. But if you are talking about false positives, then you have to refer to markers and their associations.
[Original] 28. Page 12, lines 327 to 328: it should be “phenotypic variation. This SNP has a negative major allele effect of -1.31 and a positive minor allele effect of 0.44 for MLW.”.
[A Response]We have revised in the text (Results section, line316, page12).
[R Response] Please revise again, the mistake is still there.

Experimental design

Please read the basic report.

Validity of the findings

Please read the basic report.

Additional comments

Please read the basic report.